# Accelerated Mirror Descent
# in Continuous and Discrete Time

**Walid Krichene**
UC Berkeley
walid@eecs.berkeley.edu

**Alexandre M. Bayen**
UC Berkeley
bayen@berkeley.edu

**Peter L. Bartlett**
UC Berkeley and QUT
bartlett@berkeley.edu

## Abstract

We study accelerated mirror descent dynamics in continuous and discrete time. Combining the original continuous-time motivation of mirror descent with a recent ODE interpretation of Nesterov's accelerated method, we propose a family of continuous-time descent dynamics for convex functions with Lipschitz gradients, such that the solution trajectories converge to the optimum at a $\mathcal{O}(1/t^2)$ rate. We then show that a large family of first-order accelerated methods can be obtained as a discretization of the ODE, and these methods converge at a $\mathcal{O}(1/k^2)$ rate. This connection between accelerated mirror descent and the ODE provides an intuitive approach to the design and analysis of accelerated first-order algorithms.

## 1 Introduction

We consider a convex optimization problem, minimize$_{x \in \mathcal{X}} f(x)$, where $\mathcal{X} \subseteq \mathbb{R}^n$ is convex and closed, $f$ is a $C^1$ convex function, and $\nabla f$ is assumed to be $L_f$-Lipschitz. Let $f^\star$ be the minimum of $f$ on $\mathcal{X}$. Many convex optimization methods can be interpreted as the discretization of an ordinary differential equation, the solutions of which are guaranteed to converge to the set of minimizers. Perhaps the simplest such method is gradient descent, given by the iteration $x^{(k+1)} = x^{(k)} - s\nabla f(x^{(k)})$ for some step size $s$, which can be interpreted as the discretization of the ODE $\dot{X}(t) = -\nabla f(X(t))$, with discretization step $s$. The well-established theory of ordinary differential equations can provide guidance in the design and analysis of optimization algorithms, and has been used for unconstrained optimization [8, 7, 13], constrained optimization [27] and stochastic optimization [25]. In particular, proving convergence of the solution trajectories of an ODE can often be achieved using simple and elegant Lyapunov arguments. The ODE can then be carefully discretized to obtain an optimization algorithm for which the convergence rate can be analyzed by using an analogous Lyapunov argument in discrete time.

In this article, we focus on two families of first-order methods: Nesterov's accelerated method [22], and Nemirovski's mirror descent method [19]. First-order methods have become increasingly important for large-scale optimization problems that arise in machine learning applications. Nesterov's accelerated method [22] has been applied to many problems and extended in a number of ways, see for example [23, 20, 21, 4]. The mirror descent method also provides an important generalization of the gradient descent method to non-Euclidean geometries, as discussed in [19, 3], and has many applications in convex optimization [6, 5, 12, 15], as well as online learning [9, 11]. An intuitive understanding of these methods is of particular importance for the design and analysis of new algorithms. Although Nesterov's method has been notoriously hard to explain intuitively [14], progress has been made recently: in [28], Su et al. give an ODE interpretation of Nesterov's method. However, this interpretation is restricted to the original method [22], and does not apply to its extensions to non-Euclidean geometries. In [1], Allen-Zhu and Orecchia give another interpretation of Nesterov's method, as performing, at each iteration, a convex combination of a mirror step and a gradient step. Although it covers a broader family of algorithms (including non-Euclidean geometries), this interpretation still requires an involved analysis, and lacks the simplicity and elegance of

ODEs. We provide a new interpretation which has the benefits of both approaches: we show that a broad family of accelerated methods (which includes those studied in [28] and [1]) can be obtained as a discretization of a simple ODE, which converges at a $\mathcal{O}(1/t^2)$ rate. This provides a unified interpretation, which could potentially simplify the design and analysis of first-order accelerated methods.

The continuous-time interpretation [28] of Nesterov's method and the continuous-time motivation of mirror descent [19] both rely on a Lyapunov argument. They are reviewed in Section 2. By combining these ideas, we propose, in Section 3, a candidate Lyapunov function $V(X(t), Z(t), t)$ that depends on two state variables: $X(t)$, which evolves in the primal space $E = \mathbb{R}^n$, and $Z(t)$, which evolves in the dual space $E^*$, and we design coupled dynamics of $(X, Z)$ to guarantee that $\frac{d}{dt} V(X(t), Z(t), t) \leq 0$. Such a function is said to be a Lyapunov function, in reference to [18]; see also [16]. This leads to a new family of ODE systems, given in Equation (5). We prove the existence and uniqueness of the solution to (5) in Theorem 1. Then we prove in Thereom 2, using the Lyapunov function $V$, that the solution trajectories are such that $f(X(t)) - f^\star = \mathcal{O}(1/t^2)$. In Section 4, we give a discretization of these continuous-time dynamics, and obtain a family of accelerated mirror descent methods, for which we prove the same $\mathcal{O}(1/k^2)$ convergence rate (Theorem 3) using a Lyapunov argument analogous to (though more involved than) the continuous-time case. We give, as an example, a new accelerated method on the simplex, which can be viewed as performing, at each step, a convex combination of two entropic projections with different step sizes. This ODE interpretation of accelerated mirror descent gives new insights and allows us to extend recent results such as the adaptive restarting heuristics proposed by O'Donoghue and Candès in [24], which are known to empirically improve the convergence rate. We test these methods on numerical examples in Section 5 and comment on their performance.

## 2   ODE interpretations of Nemirovski's mirror descent method and Nesterov's accelerated method

Proving convergence of the solution trajectories of an ODE often involves a Lyapunov argument. For example, to prove convergence of the solutions to the gradient descent ODE $\dot{X}(t) = -\nabla f(X(t))$, consider the Lyapunov function $V(X(t)) = \frac{1}{2}\|X(t) - x^\star\|^2$ for some minimizer $x^\star$. Then the time derivative of $V(X(t))$ is given by

$$\frac{d}{dt} V(X(t)) = \left\langle \dot{X}(t), X(t) - x^\star \right\rangle = \langle -\nabla f(X(t)), X(t) - x^\star \rangle \leq -(f(X(t)) - f^\star),$$

where the last inequality is by convexity of $f$. Integrating, we have $V(X(t)) - V(x_0) \leq tf^\star - \int_0^t f(X(\tau))d\tau$, thus by Jensen's inequality, $f\left(\frac{1}{t} \int_0^t X(\tau)d\tau\right) - f^\star \leq \frac{1}{t} \int_0^t f(X(\tau))d\tau - f^\star \leq \frac{V(x_0)}{t}$, which proves that $f\left(\frac{1}{t} \int_0^t X(\tau)d\tau\right)$ converges to $f^\star$ at a $\mathcal{O}(1/t)$ rate.

### 2.1   Mirror descent ODE

The previous argument was extended by Nemirovski and Yudin in [19] to a family of methods called mirror descent. The idea is to start from a non-negative function $V$, then to design dynamics for which $V$ is a Lyapunov function. Nemirovski and Yudin argue that one can replace the Lyapunov function $V(X(t)) = \frac{1}{2}\|X(t) - x^\star\|^2$ by a function on the dual space, $V(Z(t)) = D_{\psi^*}(Z(t), z^\star)$, where $Z(t) \in E^*$ is a dual variable for which we will design the dynamics ($z^\star$ is the value of $Z$ at equilibrium), and the corresponding trajectory in the primal space is $X(t) = \nabla \psi^*(Z(t))$. Here $\psi^*$ is a convex function defined on $E^*$, such that $\nabla \psi^*$ maps $E^*$ to $\mathcal{X}$, and $D_{\psi^*}(Z(t), z^\star)$ is the Bregman divergence associated with $\psi^*$, defined as $D_{\psi^*}(z, y) = \psi^*(z) - \psi^*(y) - \langle \nabla \psi^*(y), z - y \rangle$. The function $\psi^*$ is said to be $\ell$-strongly convex w.r.t. a reference norm $\|\cdot\|_*$ if $D_\psi^*(z, y) \geq \frac{\ell}{2}\|z - y\|_*^2$ for all $y, z$, and it is said to be $L$-smooth w.r.t. $\|\cdot\|_*$ if $D_{\psi^*}(z, y) \leq \frac{L}{2}\|z - y\|_*^2$. For a review of properties of Bregman divergences, see Chapter 11.2 in [11], or Appendix A in [2].

By definition of the Bregman divergence, we have

$$\frac{d}{dt} V(Z(t)) = \frac{d}{dt} D_{\psi^*}(Z(t), z^\star) = \frac{d}{dt} \left(\psi^*(Z(t)) - \psi^*(z^\star) - \langle \nabla \psi^*(z^\star), Z(t) - z^\star \rangle\right)$$
$$= \left\langle \nabla \psi^*(Z(t)) - \nabla \psi^*(z^\star), \dot{Z}(t) \right\rangle = \left\langle X(t) - x^\star, \dot{Z}(t) \right\rangle.$$

Therefore, if the dual variable $Z$ obeys the dynamics $\dot{Z} = -\nabla f(X)$, then

$$\frac{d}{dt}V(Z(t)) = -\langle \nabla f(X(t)), X(t) - x^\star \rangle \leq -(f(X(t)) - f^\star)$$

and by the same argument as in the gradient descent ODE, $V$ is a Lyapunov function and $f\left(\frac{1}{t}\int_0^t X(\tau)d\tau\right) - f^\star$ converges to 0 at a $\mathcal{O}(1/t)$ rate. The mirror descent ODE system can be summarized by

$$\begin{cases} X = \nabla\psi^*(Z) \\ \dot{Z} = -\nabla f(X) \\ X(0) = x_0, \ Z(0) = z_0 \text{ with } \nabla\psi^*(z_0) = x_0 \end{cases} \tag{1}$$

Note that since $\nabla\psi^*$ maps into $\mathcal{X}$, $X(t) = \nabla\psi^*(Z(t))$ remains in $\mathcal{X}$. Finally, the unconstrained gradient descent ODE can be obtained as a special case of the mirror descent ODE (1) by taking $\psi^*(z) = \frac{1}{2}\|z\|^2$, for which $\nabla\psi^*$ is the identity, in which case $X$ and $Z$ coincide.

## 2.2 ODE interpretation of Nesterov's accelerated method

In [28], Su et al. show that Nesterov's accelerated method [22] can be interpreted as a discretization of a second-order differential equation, given by

$$\begin{cases} \ddot{X} + \frac{r+1}{t}\dot{X} + \nabla f(X) = 0 \\ X(0) = x_0, \ \dot{X}(0) = 0 \end{cases} \tag{2}$$

The argument uses the following Lyapunov function (up to reparameterization), $\mathcal{E}(t) = \frac{t^2}{r}(f(X) - f^\star) + \frac{r}{2}\|X + \frac{t}{r}\dot{X} - x^\star\|^2$, which is proved to be a Lyapunov function for the ODE (2) whenever $r \geq 2$. Since $\mathcal{E}$ is decreasing along trajectories of the system, it follows that for all $t > 0$, $\mathcal{E}(t) \leq \mathcal{E}(0) = \frac{r}{2}\|x_0 - x^\star\|^2$, therefore $f(X(t)) - f^\star \leq \frac{r}{t^2}\mathcal{E}(t) \leq \frac{r}{t^2}\mathcal{E}(0) \leq \frac{r^2}{t^2}\frac{\|x_0 - x^\star\|^2}{2}$, which proves that $f(X(t))$ converges to $f^\star$ at a $\mathcal{O}(1/t^2)$ rate. One should note in particular that the squared Euclidean norm is used in the definition of $E(t)$ and, as a consequence, discretizing the ODE (2) leads to a family of unconstrained, Euclidean accelerated methods. In the next section, we show that by combining this argument with Nemirovski's idea of using a general Bregman divergence as a Lyapunov function, we can construct a much more general family of ODE systems which have the same $\mathcal{O}(1/t^2)$ convergence guarantee. And by discretizing the resulting dynamics, we obtain a general family of accelerated methods that are not restricted to the unconstrained Euclidean geometry.

# 3 Continuous-time Accelerated Mirror Descent

## 3.1 Derivation of the accelerated mirror descent ODE

We consider a pair of dual convex functions, $\psi$ defined on $\mathcal{X}$ and $\psi^*$ defined on $E^*$, such that $\nabla\psi^* : E^* \to \mathcal{X}$. We assume that $\psi^*$ is $L_{\psi^*}$-smooth with respect to $\|\cdot\|_*$, a reference norm on the dual space. Consider the function

$$V(X(t), Z(t), t) = \frac{t^2}{r}(f(X(t)) - f^\star) + rD_{\psi^*}(Z(t), z^\star) \tag{3}$$

where $Z$ is a dual variable for which we will design the dynamics, and $z^\star$ is its value at equilibrium. Taking the time-derivative of $V$, we have

$$\frac{d}{dt}V(X(t), Z(t), t) = \frac{2t}{r}(f(X) - f^\star) + \frac{t^2}{r}\left\langle \nabla f(X), \dot{X} \right\rangle + r\left\langle \dot{Z}, \nabla\psi^*(Z) - \nabla\psi^*(z^\star) \right\rangle.$$

Assume that $\dot{Z} = -\frac{t}{r}\nabla f(X)$. Then, the time-derivative of $V$ becomes

$$\frac{d}{dt}V(X(t), Z(t), t) = \frac{2t}{r}(f(X) - f^\star) - t\left\langle \nabla f(X), -\frac{t}{r}\dot{X} + \nabla\psi^*(Z) - \nabla\psi^*(z^\star) \right\rangle.$$

Therefore, if $Z$ is such that $\nabla\psi^*(Z) = X + \frac{t}{r}\dot{X}$, and $\nabla\psi^*(z^\star) = x^\star$, then,

$$\frac{d}{dt}V(X(t), Z(t), t) = \frac{2t}{r}(f(X) - f^\star) - t\langle \nabla f(X), X - x^\star \rangle \leq \frac{2t}{r}(f(X) - f^\star) - t(f(X) - f^\star)$$

$$\leq -t\frac{r-2}{r}(f(X) - f^\star) \tag{4}$$

and it follows that $V$ is a Lyapunov function whenever $r \geq 2$. The proposed ODE system is then

$$\begin{cases} \dot{X} = \frac{r}{t}(\nabla\psi^*(Z) - X), \\ \dot{Z} = -\frac{t}{r}\nabla f(X), \\ X(0) = x_0, \ Z(0) = z_0, \ \text{with } \nabla\psi^*(z_0) = x_0. \end{cases} \tag{5}$$

In the unconstrained Euclidean case, taking $\psi^*(z) = \frac{1}{2}\|z\|^2$, we have $\nabla\psi^*(z) = z$, thus $Z = X + \frac{t}{r}\dot{X}$, and the ODE system is equivalent to $\frac{d}{dt}\left(X + \frac{t}{r}\dot{X}\right) = -\frac{t}{r}\nabla f(X)$, which is equivalent to the ODE (2) studied in [28], which we recover as a special case.

We also give another interpretation of ODE (5): the first equation is equivalent to $t^r \dot{X} + rt^{r-1}X = rt^{r-1}\nabla\psi^*(Z)$, or, in integral form, $t^r X(t) = r\int_0^t \tau^{r-1}\nabla\psi^*(Z(\tau))d\tau$, which can be written as $X(t) = \frac{\int_0^t w(\tau)\nabla\psi^*(Z(\tau))d\tau}{\int_0^t w(\tau)d\tau}$, with $w(\tau) = \tau^{r-1}$. Therefore the coupled dynamics of $(X, Z)$ can be interpreted as follows: the dual variable $Z$ accumulates gradients with a $\frac{t}{r}$ rate, while the primal variable $X$ is a weighted average of $\nabla\psi^*(Z(\tau))$ (the "mirrored" dual trajectory), with weights proportional to $\tau^{r-1}$. This also gives an interpretation of $r$ as a parameter controlling the weight distribution. It is also interesting to observe that the weights are increasing if and only if $r \geq 2$. Finally, with this averaging interpretation, it becomes clear that the primal trajectory $X(t)$ remains in $\mathcal{X}$, since $\nabla\psi^*$ maps into $\mathcal{X}$ and $\mathcal{X}$ is convex.

## 3.2 Solution of the proposed dynamics

First, we prove existence and uniqueness of a solution to the ODE system (5), defined for all $t > 0$. By assumption, $\psi^*$ is $L_{\psi^*}$-smooth w.r.t. $\|\cdot\|_*$, which is equivalent (see e.g. [26]) to $\nabla\psi^*$ is $L_{\psi^*}$-Lipschitz. Unfortunately, due to the $\frac{r}{t}$ term in the expression of $\dot{X}$, the function $(X, Z, t) \mapsto (\dot{X}, \dot{Z})$ is not Lipschitz at $t = 0$, and we cannot directly apply the Cauchy-Lipschitz existence and uniqueness theorem. However, one can work around it by considering a sequence of approximating ODEs, similarly to the argument used in [28].

**Theorem 1.** *Suppose $f$ is $C^1$, and that $\nabla f$ is $L_f$-Lipschitz, and let $(x_0, z_0) \in \mathcal{X} \times E^*$ such that $\nabla\psi^*(z_0) = x_0$. Then the accelerated mirror descent ODE system (5) with initial condition $(x_0, z_0)$ has a unique solution $(X, Z)$, in $C^1([0, \infty), \mathbb{R}^n)$.*

We will show existence of a solution on any given interval $[0, T]$ (uniqueness is proved in the supplementary material). Let $\delta > 0$, and consider the smoothed ODE system

$$\begin{cases} \dot{X} = \frac{r}{\max(t,\delta)}(\nabla\psi^*(Z) - X), \\ \dot{Z} = -\frac{t}{r}\nabla f(X), \\ X(0) = x_0, \ Z(0) = z_0 \ \text{with } \nabla\psi^*(z_0) = x_0. \end{cases} \tag{6}$$

Since the functions $(X, Z) \mapsto -\frac{t}{r}\nabla f(X)$ and $(X, Z) \mapsto \frac{r}{\max(t,\delta)}(\nabla\psi^*(Z) - X)$ are Lipschitz for all $t \in [0, T]$, by the Cauchy-Lipschitz theorem (Theorem 2.5 in [29]), the system (6) has a unique solution $(X_\delta, Z_\delta)$ in $C^1([0, T])$. In order to show the existence of a solution to the original ODE, we use the following Lemma (proved in the supplementary material).

**Lemma 1.** *Let $t_0 = \frac{2}{\sqrt{L_f L_{\psi^*}}}$. Then the family of solutions $\left((X_\delta, Z_\delta)|_{[0,t_0]}\right)_{\delta \leq t_0}$ is equi-Lipschitz-continuous and uniformly bounded.*

*Proof of existence.* Consider the family of solutions $\left((X_{\delta_i}, Z_{\delta_i}), \delta_i = t_0 2^{-i}\right)_{i \in \mathbb{N}}$ restricted to $[0, t_0]$. By Lemma 1, this family is equi-Lipschitz-continuous and uniformly bounded, thus by the Arzelà-Ascoli theorem, there exists a subsequence $\left((X_{\delta_i}, Z_{\delta_i})\right)_{i \in \mathcal{I}}$ that converges uniformly on $[0, t_0]$ (where $\mathcal{I} \subset \mathbb{N}$ is an infinite set of indices). Let $(\bar{X}, \bar{Z})$ be its limit. Then we prove that $(\bar{X}, \bar{Z})$ is a solution to the original ODE (5) on $[0, t_0]$.

First, since for all $i \in \mathcal{I}$, $X_{\delta_i}(0) = x_0$ and $Z_{\delta_i}(0) = z_0$, it follows that $\bar{X}(0) = \lim_{i\to\infty, i\in\mathcal{I}} X_{\delta_i}(0) = x_0$ and $\bar{Z}(0) = \lim_{i\to\infty, i\in\mathcal{I}} Z_{\delta_i}(0) = z_0$, thus $(\bar{X}, \bar{Z})$ satisfies the initial conditions. Next, let $t_1 \in (0, t_0)$, and let $(\tilde{X}, \tilde{Z})$ be the solution of the ODE (5) on $t \geq t_1$, with initial condition $(\bar{X}(t_1), \bar{Z}(t_1))$. Since $(X_{\delta_i}(t_1), Z_{\delta_i}(t_1))_{i\in\mathcal{I}} \to (\bar{X}(t_1), \bar{Z}(t_1))$ as $i \to \infty$, then by

continuity of the solution w.r.t. initial conditions (Theorem 2.8 in [29]), we have that for some $\epsilon > 0$, $X_{\delta_i} \to \tilde{X}$ uniformly on $[t_1, t_1 + \epsilon]$. But we also have $X_{\delta_i} \to \bar{X}$ uniformly on $[0, t_0]$, therefore $\bar{X}$ and $\tilde{X}$ coincide on $[t_1, t_1 + \epsilon)$, therefore $\bar{X}$ satisfies the ODE on $[t_1, t_1 + \epsilon)$. And since $t_1$ is arbitrary in $(0, t_0)$, this concludes the proof of existence. $\square$

### 3.3 Convergence rate

It is now straightforward to establish the convergence rate of the solution.

**Theorem 2.** *Suppose that $f$ has Lipschitz gradient, and that $\psi^*$ is a smooth distance generating function. Let $(X(t), Z(t))$ be the solution to the accelerated mirror descent ODE* (5) *with $r \geq 2$. Then for all $t > 0$, $f(X(t)) - f^\star \leq \frac{r^2 D_{\psi^*}(z_0, z^\star)}{t^2}$.*

*Proof.* By construction of the ODE, $V(X(t), Z(t), t) = \frac{t^2}{r}(f(X(t)) - f^\star) + r D_{\psi^*}(Z(t), z^\star)$ is a Lyapunov function. It follows that for all $t > 0$, $\frac{t^2}{r}(f(X(t)) - f^\star) \leq V(X(t), Z(t), t) \leq V(x_0, z_0, 0) = r D_{\psi^*}(z_0, z^\star)$. $\square$

## 4 Discretization

Next, we show that with a careful discretization of this continuous-time dynamics, we can obtain a general family of accelerated mirror descent methods for constrained optimization. Using a mixed forward/backward Euler scheme (see e.g. Chapter 2 in [10]), we can discretize the ODE system (5) using a step size $\sqrt{s}$ as follows. Given a solution $(X, Z)$ of the ODE (5), let $t_k = k\sqrt{s}$, and $x^{(k)} = X(t_k) = X(k\sqrt{s})$. Approximating $\dot{X}(t_k)$ with $\frac{X(t_k + \sqrt{s}) - X(t_k)}{\sqrt{s}}$, we propose the discretization

$$\begin{cases} \frac{x^{(k+1)} - x^{(k)}}{\sqrt{s}} = \frac{r}{k\sqrt{s}}\left(\nabla\psi^*(z^{(k)}) - x^{(k+1)}\right), \\ \frac{z^{(k+1)} - z^{(k)}}{\sqrt{s}} + \frac{k\sqrt{s}}{r}\nabla f(x^{(k+1)}) = 0. \end{cases} \tag{7}$$

The first equation can be rewritten as $x^{(k+1)} = \left(x^{(k)} + \frac{r}{k}\nabla\psi^*(z^{(k)})\right) / \left(1 + \frac{r}{k}\right)$ (note the independence on $s$, due to the time-scale invariance of the first ODE). In other words, $x^{(k+1)}$ is a convex combination of $\nabla\psi^*(z^{(k)})$ and $x^{(k)}$ with coefficients $\lambda_k = \frac{r}{r+k}$ and $1 - \lambda_k = \frac{k}{r+k}$. To summarize, our first discrete scheme can be written as

$$\begin{cases} x^{(k+1)} = \lambda_k \nabla\psi^*(z^{(k)}) + (1 - \lambda_k)x^{(k)}, \ \lambda_k = \frac{r}{r+k}, \\ z^{(k+1)} = z^{(k)} - \frac{ks}{r}\nabla f(x^{(k+1)}). \end{cases} \tag{8}$$

Since $\nabla\psi^*$ maps into the feasible set $\mathcal{X}$, starting from $x^{(0)} \in \mathcal{X}$ guarantees that $x^{(k)}$ remains in $\mathcal{X}$ for all $k$ (by convexity of $\mathcal{X}$). Note that by duality, we have $\nabla\psi^*(x^*) = \arg\max_{x \in \mathcal{X}} \langle x, x^* \rangle - \psi(x)$, and if we additionally assume that $\psi$ is differentiable on the image of $\nabla\psi^*$, then $\nabla\psi = (\nabla\psi^*)^{-1}$ (Theorem 23.5 in [26]), thus if we write $\tilde{z}^{(k)} = \nabla\psi^*(z^{(k)})$, the second equation can be written as

$$\tilde{z}^{(k+1)} = \nabla\psi^*(\nabla\psi(\tilde{z}^{(k)}) - \frac{ks}{r}\nabla f(x^{(k+1)})) = \arg\min_{x \in \mathcal{X}} \psi(x) - \left\langle \nabla\psi(\tilde{z}^{(k)}) - \frac{ks}{r}\nabla f(x^{(k+1)}), x \right\rangle$$

$$= \arg\min_{x \in \mathcal{X}} \frac{ks}{r}\left\langle \nabla f(x^{(k+1)}), x \right\rangle + D_{\psi}(x, \tilde{z}^{(k)}).$$

We will eventually modify this scheme in order to be able to prove the desired $\mathcal{O}(1/k^2)$ convergence rate. However, we start by analyzing this version. Motivated by the continuous-time Lyapunov function (3), and using the correspondence $t \approx k\sqrt{s}$, we consider the potential function $E^{(k)} = V(x^{(k)}, z^{(k)}, k\sqrt{s}) = \frac{k^2 s}{r}(f(x^{(k)}) - f^\star) + r D_{\psi^*}(z^{(k)}, z^\star)$. Then we have

$$E^{(k+1)} - E^{(k)} = \frac{(k+1)^2 s}{r}(f(x^{(k+1)}) - f^\star) - \frac{k^2 s}{r}(f(x^{(k)}) - f^\star) + r(D_{\psi^*}(z^{(k+1)}, z^\star) - D_{\psi^*}(z^{(k)}, z^\star))$$

$$= \frac{k^2 s}{r}(f(x^{(k+1)}) - f(x^{(k)})) + \frac{s(1+2k)}{r}(f(x^{(k+1)}) - f^\star) + r(D_{\psi^*}(z^{(k+1)}, z^\star) - D_{\psi^*}(z^{(k)}, z^\star)).$$

And through simple algebraic manipulation, the last term can be bounded as follows

$$D_{\psi^*}(z^{(k+1)}, z^\star) - D_{\psi^*}(z^{(k)}, z^\star)$$

$$= D_{\psi^*}(z^{(k+1)}, z^{(k)}) + \left\langle \nabla\psi^*(z^{(k)}) - \nabla\psi^*(z^\star), z^{(k+1)} - z^{(k)} \right\rangle \quad \text{by definition of the Bregman divergence}$$

$$= D_{\psi^*}(z^{(k+1)}, z^{(k)}) + \left\langle \frac{k}{r}(x^{(k+1)} - x^{(k)}) + x^{(k+1)} - x^\star, -\frac{ks}{r}\nabla f(x^{(k+1)}) \right\rangle \quad \text{by the discretization (8)}$$

$$\leq D_{\psi^*}(z^{(k+1)}, z^{(k)}) + \frac{k^2 s}{r^2}(f(x^{(k)}) - f(x^{(k+1)})) + \frac{ks}{r}(f^\star - f(x^{(k+1)})). \quad \text{by convexity of } f$$

Therefore we have $E^{(k+1)} - E^{(k)} \leq -\frac{s[(r-2)k-1]}{r}(f(x^{(k+1)}) - f^\star) + rD_{\psi^*}(z^{(k+1)}, z^{(k)})$. Comparing this expression with the expression (4) of $\frac{d}{dt}V(X(t), Z(t), t)$ in the continuous-time case, we see that we obtain an analogous expression, except for the additional Bregman divergence term $rD_{\psi^*}(z^{(k+1)}, z^{(k)})$, and we cannot immediately conclude that $V$ is a Lyapunov function. This can be remedied by the following modification of the discretization scheme.

## 4.1 A family of discrete-time accelerated mirror descent methods

In the expression (8) of $x^{(k+1)} = \lambda_k \tilde{z}^{(k)} + (1 - \lambda_k)x^{(k)}$, we propose to replace $x^{(k)}$ with $\tilde{x}^{(k)}$, obtained as a solution to a minimization problem $\tilde{x}^{(k)} = \arg\min_{x \in \mathcal{X}} \gamma s \left\langle \nabla f(x^{(k)}), x \right\rangle + R(x, x^{(k)})$, where $R$ is regularization function that satisfies the following assumptions: there exist $0 < \ell_R \leq L_R$ such that for all $x, x' \in \mathcal{X}$, $\frac{\ell_R}{2}\|x - x'\|^2 \leq R(x, x') \leq \frac{L_R}{2}\|x - x'\|^2$.

In the Euclidean case, one can take $R(x, x') = \frac{\|x-x'\|^2}{2}$, in which case $\ell_R = L_R = 1$ and the $\tilde{x}$ update becomes a prox-update. In the general case, one can take $R(x, x') = D_\phi(x, x')$ for some distance generating function $\phi$ which is $\ell_R$-strongly convex and $L_R$-smooth, in which case the $\tilde{x}$ update becomes a mirror update. The resulting method is summarized in Algorithm 1. This algorithm is a generalization of Allen-Zhu and Orecchia's interpretation of Nesterov's method in [1], where $x^{(k+1)}$ is a convex combination of a mirror descent update and a gradient descent update.

---

**Algorithm 1** Accelerated mirror descent with distance generating function $\psi^*$, regularizer $R$, step size $s$, and parameter $r \geq 3$

---

1: Initialize $\tilde{x}^{(0)} = x_0$, $\tilde{z}^{(0)} = x_0$, $\left(\text{or } z^{(0)} \in (\nabla\psi)^{-1}(x_0)\right)$.
2: **for** $k \in \mathbb{N}$ **do**
3: $\quad x^{(k+1)} = \lambda_k \tilde{z}^{(k)} + (1 - \lambda_k)\tilde{x}^{(k)}$, with $\lambda_k = \frac{r}{r+k}$.
4: $\quad \tilde{z}^{(k+1)} = \arg\min_{\tilde{z} \in \mathcal{X}} \frac{ks}{r}\left\langle \nabla f(x^{(k+1)}), \tilde{z} \right\rangle + D_\psi(\tilde{z}, \tilde{z}^{(k)})$.
$\quad\quad \left(\text{If } \psi \text{ is non-differentiable, } z^{(k+1)} = z^{(k)} - \frac{kr}{s}\nabla f(x^{(k+1)}) \text{ and } \tilde{z}^{(k+1)} = \nabla\psi^*(z^{(k+1)}).\right)$
5: $\quad \tilde{x}^{(k+1)} = \arg\min_{\tilde{x} \in \mathcal{X}} \gamma s \left\langle \nabla f(x^{(k+1)}), \tilde{x} \right\rangle + R(\tilde{x}, x^{(k+1)})$

---

## 4.2 Consistency of the modified scheme

One can show that given our assumptions on $R$, $\tilde{x}^{(k)} = x^{(k)} + \mathcal{O}(s)$. Indeed, we have

$$\frac{\ell_R}{2}\|\tilde{x}^{(k)} - x^{(k)}\|^2 \leq R(\tilde{x}^{(k)}, x^{(k)}) \leq R(x^{(k)}, x^{(k)}) + \gamma s \left\langle \nabla f(x^{(k)}), x^{(k)} - \tilde{x}^{(k)} \right\rangle$$

$$\leq \gamma s \|\nabla f(x^{(k)})\|_* \|\tilde{x}^{(k)} - x^{(k)}\|$$

therefore $\|\tilde{x}^{(k)} - x^{(k)}\| \leq s\frac{2\gamma\|\nabla f(x^{(k)})\|_*}{\ell_R}$, which proves the claim. Using this observation, we can show that the modified discretization scheme is consistent with the original ODE (5), that is, the difference equations defining $x^{(k)}$ and $z^{(k)}$ converge, as $s$ tends to 0, to the ordinary differential equations of the continuous-time system (5). The difference equations of Algorithm 1 are equivalent to (7) in which $x^{(k)}$ is replaced by $\tilde{x}^{(k)}$, i.e.

$$\begin{cases} \frac{x^{(k+1)} - \tilde{x}^{(k)}}{\sqrt{s}} = \frac{r}{k\sqrt{s}}(\nabla\psi^*(z^{(k)}) - x^{(k+1)}) \\ \frac{z^{(k+1)} - z^{(k)}}{\sqrt{s}} = -\frac{k\sqrt{s}}{r}\nabla f(x^{(k+1)}) \end{cases}$$

Now suppose there exist $C^1$ functions $(X, Z)$, defined on $\mathbb{R}_+$, such that $X(t_k) \approx x^{(k)}$ and $Z(t_k) \approx z^{(k)}$ for $t_k = k\sqrt{s}$. Then, using the fact that $\tilde{x}^{(k)} = x^{(k)} + \mathcal{O}(s)$, we have $\frac{x^{(k+1)} - \tilde{x}^{(k)}}{\sqrt{s}} = \frac{x^{(k+1)} - x^{(k)}}{\sqrt{s}} + \mathcal{O}(\sqrt{s}) \approx \frac{X(t_k + \sqrt{s}) - X(t_k)}{\sqrt{s}} + \mathcal{O}(\sqrt{s}) = \dot{X}(t_k) + o(1)$, and similarly, $\frac{z^{(k+1)} - z^{(k)}}{\sqrt{s}} \approx \dot{Z}(t_k) + o(1)$, therefore the difference equation system can be written as

$$
\begin{cases}
\dot{X}(t_k) + o(1) = \frac{r}{t_k}(\nabla \psi^*(Z(t_k)) - X(t_k + \sqrt{s})) \\
\dot{Z}(t_k) + o(1) = -\frac{t_k}{r}\nabla f(X(t_k + \sqrt{s}))
\end{cases}
$$

which converges to the ODE (5) as $s \to 0$.

### 4.3 Convergence rate

To prove convergence of the algorithm, consider the modified potential function

$$
\tilde{E}^{(k)} = V(\tilde{x}^{(k)}, z^{(k)}, k\sqrt{s}) = \frac{k^2 s}{r}(f(\tilde{x}^{(k)}) - f^\star) + r D_{\psi^*}(z^{(k)}, z^\star).
$$

**Lemma 2.** *If $\gamma \geq L_R L_{\psi^*}$ and $s \leq \frac{\ell_R}{2L_f \gamma}$, then for all $k \geq 0$,*

$$
\tilde{E}^{(k+1)} - \tilde{E}^{(k)} \leq \frac{(2k + 1 - kr)s}{r}(f(\tilde{x}^{(k+1)}) - f^\star).
$$

*As a consequence, if $r \geq 3$, $\tilde{E}$ is a Lyapunov function for $k \geq 1$.*

This lemma is proved in the supplementary material.

**Theorem 3.** *The discrete-time accelerated mirror descent Algorithm 1 with parameter $r \geq 3$ and step sizes $\gamma \geq L_R L_{\psi^*}$, $s \leq \frac{\ell_R}{2L_f \gamma}$, guarantees that for all $k > 0$,*

$$
f(\tilde{x}^{(k)}) - f^\star \leq \frac{r}{sk^2}\tilde{E}^{(1)} \leq \frac{r^2 D_{\psi^*}(z_0, z^\star)}{sk^2} + \frac{f(x_0) - f^\star}{k^2}.
$$

*Proof.* The first inequality follows immediately from Lemma 2. The second inequality follows from a simple bound on $\tilde{E}^{(1)}$, proved in the supplementary material. □

### 4.4 Example: accelerated entropic descent

We give an instance of Algorithm 1 for simplex-constrained problems. Suppose that $\mathcal{X} = \Delta^n = \{x \in \mathbb{R}^n_+ : \sum_{i=1}^n x_i = 1\}$ is the $n$-simplex. Taking $\psi$ to be the negative entropy on $\Delta$, we have for $x \in \mathcal{X}, z \in E^*$,

$$
\psi(x) = \sum_{i=1}^n x_i \ln x_i + \delta(x|\Delta), \ \psi^*(z) = \ln\left(\sum_{i=1}^n e^{z_i}\right), \ \partial\psi(x) = (1 + \ln x_i)_i + \mathbb{R}u, \ \nabla\psi^*(z)_i = \frac{e^{z_i}}{\sum_{j=1}^n e^{z_j}}.
$$

where $\delta(\cdot|\Delta)$ is the indicator function of the simplex ($\delta(x|\Delta) = 0$ if $x \in \Delta$ and $+\infty$ otherwise), and $u \in \mathbb{R}^n$ is a normal vector to the affine hull of the simplex. The resulting mirror descent update is a simple entropy projection and can be computed exactly in $\mathcal{O}(n)$ operations, and $\psi^*$ can be shown to be 1-smooth w.r.t. $\|\cdot\|_\infty$, see for example [3, 6]. For the second update, we take $R(x, y) = D_\phi(x, y)$ where $\phi$ is a smoothed negative entropy function defined as follows: let $\epsilon > 0$, and let $\phi(x) = \epsilon \sum_{i=1}^n (x_i + \epsilon)\ln(x_i + \epsilon) + \delta(x|\Delta)$. Although no simple, closed-form expression is known for $\nabla\phi^*$, it can be computed efficiently, in $\mathcal{O}(n \log n)$ time using a deterministic algorithm, or $\mathcal{O}(n)$ expected time using a randomized algorithm, see [17]. Additionally, $\phi$ satisfies our assumptions: it is $\frac{\epsilon}{1+n\epsilon}$-strongly convex and 1-smooth w.r.t. $\|\cdot\|_\infty$. The resulting accelerated mirror descent method on the simplex can then be implemented efficiently, and by Theorem 3 it is guaranteed to converge in $\mathcal{O}(1/k^2)$ whenever $\gamma \geq 1$ and $s \leq \frac{\epsilon}{2(1+n\epsilon)L_f \gamma}$.

## 5 Numerical Experiments

We test the accelerated mirror descent method in Algorithm 1, on simplex-constrained problems in $\mathbb{R}^n$, $n = 100$, with two different objective functions: a simple quadratic $f(x) = \langle x - x^\star, Q(x - x^\star)\rangle$, for a random positive semi-definite matrix $Q$, and a log-sum-exp function

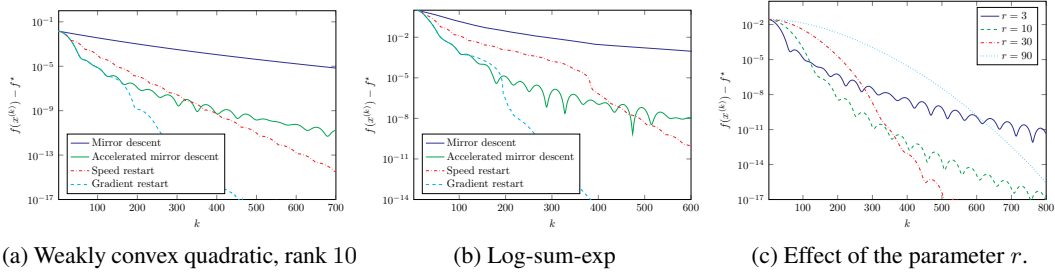

(a) Weakly convex quadratic, rank 10     (b) Log-sum-exp     (c) Effect of the parameter $r$.

Figure 1: Evolution of $f(x^{(k)}) - f^\star$ on simplex-constrained problems, using different accelerated mirror descent methods with entropy distance generating functions.

---

**Algorithm 2** Accelerated mirror descent with restart

---

1: Initialize $l = 0$, $\tilde{x}^{(0)} = \tilde{z}^{(0)} = x_0$.
2: **for** $k \in \mathbb{N}$ **do**
3:     $x^{(k+1)} = \lambda_l \tilde{z}^{(k)} + (1 - \lambda_l)\tilde{x}^{(k)}$, with $\lambda_l = \frac{r}{r+l}$
4:     $\tilde{z}^{(k+1)} = \arg\min_{\tilde{z} \in \mathcal{X}} \frac{ks}{r} \left\langle \nabla f(x^{(k+1)}), \tilde{z} \right\rangle + D_\psi(\tilde{z}, \tilde{z}^{(k)})$
5:     $\tilde{x}^{(k+1)} = \arg\min_{\tilde{x} \in \mathcal{X}} \gamma s \left\langle \nabla f(x^{(k+1)}), \tilde{x} \right\rangle + R(\tilde{x}, x^{(k+1)})$
6:     $l \leftarrow l + 1$
7:     **if** Restart condition **then**
8:        $\tilde{z}^{(k+1)} \leftarrow x^{(k+1)}$, $l \leftarrow 0$

---

given by $f(x) = \ln\left(\sum_{i=1}^{I} \langle a_i, x \rangle + b_i\right)$, where each entry in $a_i \in \mathbb{R}^n$ and $b_i \in \mathbb{R}$ is iid normal. We implement the accelerated entropic descent algorithm proposed in Section 4.4, and include the (non-accelerated) entropic descent for reference. We also adapt the gradient restarting heuristic proposed by O'Donoghue and Candès in [24], as well as the speed restart heuristic proposed by Su et al. in [28]. The generic restart method is given in Algorithm 2. The restart conditions are the following: (i) gradient restart: $\left\langle x^{(k+1)} - x^{(k)}, \nabla f(x^{(k)}) \right\rangle > 0$, and (ii) speed restart: $\|x^{(k+1)} - x^{(k)}\| < \|x^{(k)} - x^{(k-1)}\|$.

The results are given in Figure 1. The accelerated mirror descent method exhibits a polynomial convergence rate, which is empirically faster than the $\mathcal{O}(1/k^2)$ rate predicted by Theorem 3. The method also exhibits oscillations around the set of minimizers, and increasing the parameter $r$ seems to reduce the period of the oscillations, and results in a trajectory that is initially slower, but faster for large $k$, see Figure 1-c. The restarting heuristics alleviate the oscillation and empirically speed up the convergence. We also visualized, for each experiment, the trajectory of the iterates $x^{(k)}$ for each method, projected on a 2-dimensional hyperplane. The corresponding videos are included in the supplementary material.

# 6 Conclusion

By combining the Lyapunov argument that motivated mirror descent, and the recent ODE interpretation [28] of Nesterov's method, we proposed a family of ODE systems for minimizing convex functions with a Lipschitz gradient, which are guaranteed to converge at a $\mathcal{O}(1/t^2)$ rate, and proved existence and uniqueness of a solution. Then by discretizing the ODE, we proposed a family of accelerated mirror descent methods for constrained optimization and proved an analogous $\mathcal{O}(1/k^2)$ rate when the step size is small enough. The connection with the continuous-time dynamics motivates a more detailed study of the ODE (5), such as studying the oscillatory behavior of its solution trajectories, its convergence rates under additional assumptions such as strong convexity, and a rigorous study of the restart heuristics.

**Acknowledgments**

We gratefully acknowledge the NSF (CCF-1115788, CNS-1238959, CNS-1238962, CNS-1239054, CNS-1239166), the ARC (FL110100281 and ACEMS), and the Simons Institute Fall 2014 Algorithmic Spectral Graph Theory Program.

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
