[Supplementary Material · NIPS-AMD-supplementary.pdf]

# Accelerated Mirror Descent
# In Continuous and Discrete Time
# Supplementary material, NIPS 2015

Walid Krichene        Alexandre Bayen        Peter Bartlett

## 1    Mirror operator $\nabla \psi^*$

In this section, we discuss properties of distance generating functions and their subdifferentials. Let $\psi$ be a proper, closed, convex function, and suppose that $\mathcal{X}$ is the effective domain of $\psi$ (i.e. $\mathcal{X} = \{x \in \mathbb{R}^n : \psi(x) < \infty\}$). The subdifferential of $\psi$ at $x \in \mathcal{X}$ is

$$\partial \psi(x) = \{z \in E^* : \psi(y) - \psi(x) - \langle z, y - x \rangle \geq 0 \ \forall y \in \mathcal{X}\}.$$

The domain of $\partial \psi$ is $\{x \in \mathcal{X} : \partial \psi(x) \neq \emptyset\}$.

The conjugate of $\psi$ is defined as

$$\psi^*(z) = \sup_{x \in \mathcal{X}} \langle z, x \rangle - \psi(x).$$

By Theorem 12.2 in Rockafellar [1970], $\psi^*$ is convex, closed and proper. By Theorem 23.5, we have that $\partial \psi^*$ and $\partial \psi$ are inverses of each other (in the set valued sense), and

$$\partial \psi^*(z) = \arg\max_{x \in \mathcal{X}} \langle x, z \rangle - \psi(x),$$

so $\partial \psi^*$ naturally maps into $\mathcal{X}$. The following lemma gives sufficient conditions for this mirror operator to be defined on the entire dual space $E^*$, and single valued (in other words, $\psi^*$ is finite and differentiable everywhere).

**Proposition 1.** *Let $\psi$ and its conjugate $\psi^*$ be closed proper convex functions, such that the effective domain of $\psi$ is $\mathcal{X}$. Suppose that*

(i) *$\psi$ is co-finite, that is, the epigraph of $\psi$ contains no non-vertical half-lines. (An equivalent condition is that the recession function of $\psi$ is the indicator of $0$.)*

(ii) *$\psi$ is essentially strictly convex, that is, $\psi$ is strictly convex on any convex subset of the domain of $\partial \psi$.*

*Then $\psi^*$ is finite and differentiable on $E^*$, and $\nabla \psi^*$ maps $E^*$ into $\mathcal{X}$ via the following expression: for $z \in E^*$,*

$$\nabla \psi^*(z) = \arg\max_{x \in \mathcal{X}} \langle z, x \rangle - \psi(x).$$

*Proof.* Since $\psi$ is cofinite, by Theorem 13.3 in Rockafellar [1970], $\psi^*$ is finite everywhere (domain of $\psi^*$ is $E^* = \mathbb{R}^n$). And since $\psi$ is essentially strongly convex, by Theorem 25.3 in Rockafellar [1970], $\psi^*$ is essentially smooth, and hence differentiable on the interior of its domain, which is all of $E^*$.                   $\square$

Note that $\psi$ is not necessarily differentiable: consider in particular the case where its domain $\mathcal{X}$ is contained in a hyperplane (i.e. has affine dimension at most $n - 1$), then $\psi$ is, in fact, nowhere differentiable. As a consequence, the inverse mapping of $\nabla \psi^*$, $(\nabla \psi^*)^{-1} = \partial \psi$, is not always single-valued.

## 2 Proof of Lemma 1

Let us rewrite the smoothed accelerated mirror descent ODE system

$$\begin{cases} \dot{Z} = -\frac{t}{r}\nabla f(X) \\ \dot{X} = \frac{r}{\max(t,\delta)}(\nabla\psi(Z) - X) \\ X(0) = x_0, \ Z(0) = z_0 \text{ with } \nabla\psi(z_0) = x_0. \end{cases} \tag{1}$$

By the Cauchy-Lipschitz theorem, there exists a unique solution $(X_\delta, Z_\delta)$ defined on $[0, t_{\max})$, and the solution is $C^1$. Define, for $t > 0$,

$$A_\delta(t) = \sup_{u \in [0,t]} \frac{\|\dot{Z}_\delta(u)\|}{u}$$

$$B_\delta(t) = \sup_{u \in [0,t]} \frac{\|X_\delta(u) - x_0\|}{u}$$

$$C_\delta(t) = \sup_{u \in [0,t]} \|\dot{X}_\delta(u)\|$$

These quantities are finite for the following reasons:

- $\frac{\|X_\delta(u) - x_0\|}{u} = \|\dot{X}_\delta(0)\| + o(1)$ near 0, thus $B_\delta$ is finite.

- $\|\dot{X}_\delta\|$ is continuous thus bounded on $[0, t]$, thus $C_\delta$ is finite.

- Finiteness of $A_\delta$ is a consequence of the following lemma.

To prove Lemma 1, we first need the auxiliary lemma below, that provides bounds on $A_\delta, B_\delta, C_\delta$.

**Lemma 3.** *For all $t$,*

$$rA_\delta(t) \leq \|\nabla f(x_0)\| + L_f t B_\delta(t), \tag{2}$$

$$B_\delta(t) \leq \frac{L_{\psi^*} rt}{6} A_\delta(t), \tag{3}$$

$$C_\delta(t) \leq r\left(\frac{t_0 L_{\psi^*}}{2} A_\delta(t) + B_\delta(t)\right). \tag{4}$$

*Proof.* By definition of $A_\delta$ and $B_\delta$, we have

$$\|Z_\delta(t) - z_0\| \leq \int_0^t \|\dot{Z}_\delta(v)\| dv \leq A_\delta(t) \int_0^t v\, dv = \frac{t^2}{2} A_\delta(t), \tag{5}$$

$$\|X_\delta(t) - x_0\| \leq t B_\delta(t).$$

Now, from the first equation in (6), we have for all $t \leq t_0$

$$r\frac{\|\dot{Z}_\delta(t)\|}{t} = \|\nabla f(X_\delta(t))\|$$
$$\leq \|\nabla f(x_0)\| + \|\nabla f(X_\delta(t)) - \nabla f(x_0)\|$$
$$\leq \|\nabla f(x_0)\| + L_f\|X_\delta(t) - x_0\| \qquad \nabla f \text{ is } L_f\text{-Lipschitz}$$
$$\leq \|\nabla f(x_0)\| + L_f t B_\delta(t).$$

Thus,

$$rA_\delta(t) \leq \|\nabla f(x_0)\| + L_f t B_\delta(t).$$

From the second equation in (6), we have for all $t \leq \delta$,

$$e^{\frac{rt}{\delta}}\left(\dot{X}_\delta + \frac{r}{\delta}(X_\delta - x_0)\right) = \frac{r}{\delta} e^{\frac{rt}{\delta}}(\nabla\psi^*(Z_\delta) - \nabla\psi^*(z_0)),$$

i.e.,

$$\frac{d}{dt}\left((X_\delta(t) - x_0)e^{\frac{rt}{\delta}}\right) = \frac{r}{\delta} e^{\frac{rt}{\delta}}(\nabla\psi^*(Z_\delta(t)) - \nabla\psi^*(z_0)),$$

thus integrating

$$(X_\delta(t) - x_0)e^{\frac{rt}{\delta}} = \frac{r}{\delta} \int_0^t e^{\frac{rs}{\delta}} (\nabla\psi^*(Z_\delta(s)) - \nabla\psi^*(z_0))ds$$

and taking norms

$$\|X_\delta(t) - x_0\| \leq \frac{r}{\delta} \int_0^t \|\nabla\psi^*(Z_\delta(s)) - \nabla\psi^*(z_0)\|ds$$

$$\leq \frac{L_{\psi^*}r}{\delta} \int_0^t \|Z_\delta(s) - z_0\|ds \qquad\qquad \nabla\psi^* \text{ is } L_{\psi^*}\text{-Lipschitz}$$

$$\leq \frac{L_{\psi^*}r}{\delta} \int_0^t \frac{s^2}{2} A_\delta(t)ds \qquad\qquad\qquad \text{by (5)}$$

$$= \frac{L_{\psi^*}r}{\delta} A_\delta(t) \frac{t^3}{6}$$

$$\leq \frac{L_{\psi^*}rt^2}{6} A_\delta(t).$$

For $t \geq \delta$,

$$t^r \left( \dot{X}_\delta + \frac{r}{t}(X_\delta - x_0) \right) = rt^{r-1}(\nabla\psi^*(Z_\delta) - \nabla\psi^*(z_0)),$$

i.e.

$$\frac{d}{dt} \left( t^r (X_\delta(t) - x_0) \right) = rt^{r-1}(\nabla\psi^*(Z_\delta) - \nabla\psi^*(z_0)),$$

thus integrating

$$t^r (X_\delta(t) - x_0) = \int_0^t rs^{r-1}(\nabla\psi^*(Z_\delta(s)) - \nabla\psi^*(z_0))ds$$

and taking norms

$$\|X_\delta(t) - x_0\| \leq \frac{r}{t} \int_0^t \|\nabla\psi^*(Z_\delta(s)) - \nabla\psi^*(z_0)\|ds$$

$$\leq \frac{L_{\psi^*}r}{t} \int_0^t \|Z_\delta(s) - z_0\|ds \qquad\qquad \nabla\psi^* \text{ is } L_{\psi^*}\text{-Lipschitz}$$

$$\leq \frac{L_{\psi^*}r}{t} \int_0^t \frac{s^2}{2} A_\delta(t)ds \qquad\qquad\qquad \text{by (5)}$$

$$= \frac{L_{\psi^*}r}{t} A_\delta(t) \frac{t^3}{6}$$

$$= \frac{L_{\psi^*}rt^2 A_\delta(t)}{6}.$$

Dividing by $t$ and taking the supremum, we have

$$B_\delta(t) \leq \frac{L_{\psi^*}rt}{6} A_\delta(t).$$

Finally, to bound $C_\delta$, we have from the second equation in (6), for all $t \leq t_0$,

$$\|\dot{X}_\delta(t)\| = \frac{r}{\max(\delta, t)} \|\nabla\psi^*(Z_\delta(t)) - X_\delta(t)\|$$

$$\leq \frac{r}{\max(\delta, t)} (\|\nabla\psi^*(Z_\delta(t)) - \nabla\psi^*(z_0)\| + \|X_\delta(t) - x_0\|)$$

$$\leq \frac{r}{\max(\delta, t)} (L_{\psi^*}\|Z_\delta(t) - z_0\| + \|X_\delta(t) - x_0\|)$$

$$\leq \frac{r}{\max(\delta, t)} \left( \frac{t^2}{2} L_{\psi^*} A_\delta(t) + tB_\delta(t) \right)$$

$$\leq r \left( \frac{L_{\psi^*}t_0}{2} A_\delta(t) + B_\delta(t) \right),$$

which conclude the proof. □

*Proof of Lemma 1.* First, we show that $A_\delta, B_\delta, C_\delta$ are bounded on $[0, t_0]$, uniformly in $\delta$.

Combining (2) and (3), we have

$$B_\delta(t)\frac{6}{L_{\psi^*}t} \leq rA_\delta(t) \leq \|\nabla f(x_0)\| + L_f t B_\delta(t)$$

thus

$$B_\delta(t)\left(\frac{6}{L_{\psi^*}t} - L_f t\right) \leq \|\nabla f(x_0)\|.$$

And when $t \leq \alpha\sqrt{\frac{6}{L_f L_{\psi^*}}}$,

$$\frac{6}{L_{\psi^*}t} - L_f t \geq \sqrt{\frac{6L_f}{L_{\psi^*}}}\left(\frac{1}{\alpha} - \alpha\right)$$

and for $\alpha = \sqrt{\frac{2}{3}}$, $\frac{1}{\alpha} - \alpha = \frac{1}{\sqrt{6}}$, thus setting

$$t_0 = \sqrt{\frac{2}{3}}\sqrt{\frac{6}{L_f L_{\psi^*}}} = \frac{2}{\sqrt{L_f L_{\psi^*}}}$$

we have for all $t \leq t_0$, $\frac{6}{L_{\psi^*}t} - L_f t \geq \sqrt{\frac{L_f}{L_{\psi^*}}}$, and so

$$B_\delta(t_0) \leq \sqrt{\frac{L_{\psi^*}}{L_f}}\|\nabla f(x_0)\|.$$

By (2),

$$\begin{aligned}
A_\delta(t_0) &\leq \frac{1}{r}\left(\|\nabla f(x_0)\| + L_f t_0 B_\delta(t_0)\right) \\
&\leq \frac{1}{r}\left(\|\nabla f(x_0)\| + L_f \frac{2}{\sqrt{L_f L_{\psi^*}}}\|\nabla f(x_0)\|\sqrt{\frac{L_{\psi^*}}{L_f}}\right) \\
&= \frac{3}{r}\|\nabla f(x_0)\|.
\end{aligned}$$

By (4), we have

$$\begin{aligned}
C_\delta(t_0) &\leq r\left(\frac{t_0 L_{\psi^*}}{2}A_\delta(t_0) + B_\delta(t_0)\right) \\
&\leq r\left(\frac{2}{\sqrt{L_f L_{\psi^*}}}\frac{L_{\psi^*}}{2}\frac{3}{r}\|\nabla f(x_0)\| + \sqrt{\frac{L_{\psi^*}}{L_f}}\|\nabla f(x_0)\|\right) \\
&= (3 + r)\|\nabla f(x_0)\|\sqrt{\frac{L_{\psi^*}}{L_f}}
\end{aligned}$$

To conclude, we have for all $t \in [0, t_0]$

$$\begin{aligned}
\|\dot{Z}_\delta(t)\| &\leq t_0 A_\delta(t_0), \\
\|\dot{X}_\delta(t)\| &\leq C_\delta(t_0),
\end{aligned}$$

which are bounded uniformly in $\delta$, thus the family is equi-Lipschitz-continuous on $[0, t_0]$. It also follows that it is uniformly bounded on the same interval. $\square$

# 3  Proof of uniqueness of the solution

*Proof of uniqueness.* It suffices to prove uniqueness on an open neighborhood of 0, since away from 0, uniqueness is guaranteed by the Cauchy-Lipschitz theorem.

Let $(X, Z)$ and $(\bar{X}, \bar{Z})$ be two solutions of the ODE (5), and let $\Delta_Z = Z - \bar{Z}$ and $\Delta_X = X - \bar{X}$. Then $\Delta_X, \Delta_Z$ are $C^1$, and we have

$$
\begin{cases}
\dot{\Delta}_Z = -\frac{t}{r}\left(\nabla f(X) - \nabla f(\bar{X})\right) \\
\dot{\Delta}_X = \frac{r}{t}\left(\nabla \psi^*(Z) - \nabla \psi^*(\bar{Z}) - \Delta_X\right) \\
\Delta_Z(0) = \Delta_X(0) = 0
\end{cases}
$$

Let $A(t) = \sup_{[0,t]} \frac{\|\dot{\Delta}_Z(u)\|}{u}$, and $B(t) = \sup_{[0,t]} \|\Delta_X\|$. Note that $B(t)$ is finite since $\Delta_X$ is continuous on $[0, t]$. The finiteness of $A(t)$ will be established below. We have

$$
\|\dot{\Delta}_Z(t)\| = \frac{t}{r}\|\nabla f(X(t)) - \nabla f(\bar{X}(t))\| \leq \frac{L_f t}{r}\|\Delta_X(t)\| \leq \frac{L_f t}{r} B(t).
$$

Dividing by $t$ and taking the supremum, we have

$$
A(t) \leq \frac{L_f}{r} B(t). \tag{6}
$$

Next, since $\dot{\Delta}_X + \frac{r}{t}\Delta_X = \frac{r}{t}\left(\nabla \psi^*(Z) - \nabla \psi^*(\bar{Z})\right)$, we have $\frac{d}{dt}t^r \Delta_X = rt^{r-1}\left(\nabla \psi^*(Z) - \nabla \psi^*(\bar{Z})\right)$. Therefore, integrating and taking norms

$$
t^r \|\Delta_X(t)\| \leq r \int_0^t s^{r-1}\|\nabla \psi^*(Z(s)) - \nabla \psi^*(\bar{Z}(s))\| ds \leq rt^{r-1}\int_0^t L_{\psi^*}\|\Delta_Z(s)\| ds
$$

$$
\leq L_{\psi^*} rt^{r-1} A(t) \int_0^t \frac{s^2}{2} ds = \frac{L_{\psi^*} rt^{r+2} A(t)}{6},
$$

where we used the fact that $\|\Delta_Z(s)\| = \|\int_0^s \dot{\Delta}_Z(u) du\| \leq \int_0^s u A(t) du = A(t)\frac{s^2}{2}$. Dividing by $t^r$ and taking the supremum,

$$
B(t) \leq \frac{L_{\psi^*} rt^2}{6} A(t). \tag{7}
$$

Combining (6) and (7), we have $A(t) \leq \frac{L_f L_{\psi^*} t^2}{6} A(t)$, which implies that $A(t) = 0$ for $t < \sqrt{\frac{6}{L_{\psi^*} L_f}}$, which in turn implies that $B(t) = 0$. This concludes the proof. $\qquad\square$

# 4 Proof of Lemma 2

We recall the accelerated mirror descent algorithm, the definition of the potential function, and the statement of the Lemma.

---

**Algorithm 1** Accelerated mirror descent with distance generating functions $\psi^*$ and $\phi$, step size $s$, and parameter $r \geq 3$

---

1: Initialize $\tilde{x}^{(0)} = \tilde{z}^{(0)} = x_0$.
2: **for** $k \in \mathbb{N}$ **do**
3:    $x^{(k+1)} = \lambda_k \tilde{z}^{(k)} + (1 - \lambda_k)\tilde{x}^{(k)}$, with $\lambda_k = \frac{r}{r+k}$
4:    $\tilde{z}^{(k+1)} = \arg\min_{\tilde{z} \in E} \frac{ks}{r} \left\langle \nabla f(x^{(k+1)}), \tilde{z} \right\rangle + D_\psi(\tilde{z}, \tilde{z}^{(k)}) = \nabla\psi^*(\nabla\psi(\tilde{z}^{(k)}) - \frac{ks}{r}\nabla f(x^{(k+1)}))$
5:    $\tilde{x}^{(k+1)} = \arg\min_{\tilde{x} \in E} \gamma s \left\langle \nabla f(x^{(k+1)}), \tilde{x} \right\rangle + R(\tilde{x}, x^{(k+1)})$
6: **end for**

---

We consider the function

$$\tilde{E}^{(k)} = V(\tilde{x}^{(k)}, z^{(k)}, k) = \frac{k^2 s}{r}(f(\tilde{x}^{(k)}) - f^\star) + rD_{\psi^*}(z^{(k)}, z^\star).$$

**Lemma 2.** *If $\gamma \geq L_R L_{\psi^*}$ and $s \leq \frac{\ell_R}{2L_f \gamma}$, then for all $k \geq 0$,*

$$\tilde{E}^{(k+1)} - \tilde{E}^{(k)} \leq \frac{(2k+1-kr)s}{r}(f(\tilde{x}^{(k+1)}) - f^\star).$$

In what follows, $\psi^*$ is a distance generating function that is finite and differentiable throughout $E^*$, and $\nabla\psi^*$ maps $E^*$ into $\mathcal{X}$, and is supposed to be $L_{\psi^*}$–Lipschitz in the following sense: $\|\nabla\psi^*(u) - \nabla\psi^*(v)\| \leq L_{\psi^*}\|u - v\|_*$ for all $u, v \in E^*$. The dual function $\psi$ has effective domain $\mathcal{X}$ but is not necessarily differentiable. We will need the following lemmas:

**Lemma 4.** *Let $f$ be a convex function and suppose that $\nabla f$ is $L_f$-Lipschitz w.r.t. $\|\cdot\|$. Then for all $x, x', x^+$,*

$$f(x^+) \leq f(x') + \langle\nabla f(x), x^+ - x'\rangle + \frac{L_f}{2}\|x^+ - x\|^2$$

*Proof.* Since $\nabla f$ is $L_f$-Lipschitz, we have

$$f(x^+) \leq f(x) + \langle\nabla f(x), x^+ - x\rangle + \frac{L_f}{2}\|x^+ - x\|^2$$

and by convexity of $f$,

$$f(x') \geq f(x) + \langle\nabla f(x), x' - x\rangle$$

Summing the two inequalities, we obtain the result. □

**Lemma 5.** *For all $u, v, w$*

$$D_{\psi^*}(u, v) - D_{\psi^*}(w, v) = -D_{\psi^*}(w, u) + \langle\nabla\psi^*(u) - \nabla\psi^*(v), u - w\rangle$$

*Proof.* By definition of the Bregman divergence, we have

$$\begin{aligned}
&D_{\psi^*}(u, v) - D_{\psi^*}(w, v) \\
&= \psi^*(u) - \psi^*(v) - \langle\nabla\psi^*(v), u - v\rangle - (\psi^*(w) - \psi^*(v) - \langle\nabla\psi^*(v), w - v\rangle) \\
&= \psi^*(u) - \psi^*(w) - \langle\nabla\psi^*(v), u - w\rangle \\
&= -(\psi^*(w) - \psi^*(u) - \langle\nabla\psi^*(u), w - u\rangle) + \langle\nabla\psi^*(u) - \nabla\psi^*(v), u - w\rangle \\
&= -D_{\psi^*}(w, u) + \langle\nabla\psi^*(u) - \nabla\psi^*(v), u - w\rangle
\end{aligned}$$

□

**Lemma 6.** *For all $u, v \in E^*$,*

$$\frac{1}{2L_{\psi^*}}\|\tilde{u} - \tilde{v}\|^2 \leq D_{\psi^*}(u, v) \leq \frac{L_{\psi^*}}{2}\|u - v\|_*^2$$

*where $\tilde{u} = \nabla\psi^*(u)$ and $\tilde{v} = \nabla\psi^*(v)$.*

*Proof.* We have

$$\begin{aligned}
D_{\psi^*}(u, v) &= \psi^*(u) - \psi^*(v) - \langle \nabla\psi^*(v), u - v \rangle \\
&= \int_0^1 \nabla \langle \psi^*(v + t(u - v)) - \nabla\psi^*(v), u - v \rangle \, dt \\
&\leq \|u - v\|_* \int_0^1 \|\psi^*(v + t(u - v)) - \nabla\psi^*(v)\| dt \quad \text{by the Cauchy-Schwartz inequality} \\
&\leq L_{\psi^*}\|u - v\|_* \int_0^1 \|v + t(u - v) - v\|_* dt \qquad\qquad \text{since } \psi^* \text{ is } L_{\psi^*} Lipschitz \\
&= L_{\psi^*}\|u - v\|_*^2 \int_0^1 t\, dt
\end{aligned}$$

which proves the second inequality. The first inequality will be proved by dualizing this inequality. Fix $v \in E^*$ and define

$$\begin{aligned}
h(u) &= D_{\psi^*}(u, v) = \psi^*(u) - \psi^*(v) - \langle \nabla\psi^*(v), u - v \rangle, \\
d(u) &= \frac{L_{\psi^*}}{2}\|u - v\|_*^2.
\end{aligned}$$

Then by the previous inequality, $h(u) \leq d(u)$ for all $u \in E^*$, and taking duals, we have $h^*(u^*) \geq d^*(u^*)$ for all $u^*$. We now derive the duals. Let $\tilde{v} = \psi^*(v)$. Then,

$$\begin{aligned}
h^*(u^*) &= \sup_u \langle u^*, u \rangle - h(u) \\
&= \sup_u \langle u^*, u \rangle - \psi^*(u) + \psi^*(v) + \langle \tilde{v}, u - v \rangle \\
&= \psi^*(v) - \langle v, \tilde{v} \rangle + \sup_u \langle u^* + \tilde{v}, u \rangle - \psi^*(u) \\
&= \psi^*(v) - \langle v, \tilde{v} \rangle + \psi(u^* + \tilde{v})
\end{aligned}$$

and

$$\begin{aligned}
d^*(u^*) &= \sup_u \langle u^*, u \rangle - d(u) \\
&= \sup_u \langle u^*, u \rangle - \frac{L_{\psi^*}}{2}\|u - v\|_*^2 \\
&= \sup_w \langle u^*, v + w \rangle - \frac{L_{\psi^*}}{2}\|w\|_*^2 \\
&= \langle u^*, v \rangle + \sup_w \langle u^*, w \rangle - \frac{L_{\psi^*}}{2}\|w\|_*^2 \\
&= \langle u^*, v \rangle + \frac{1}{2L_{\psi^*}}\|u^*\|^2
\end{aligned}$$

where the last equality uses Cauchy-Schwartz. Therefore combining the two inequalities,

$$\psi^*(v) - \langle v, u^* + \tilde{v} \rangle + \psi(u^* + \tilde{v}) \geq \frac{1}{2L_{\psi^*}}\|u^*\|^2$$

In particular, for all $u \in E^*$, if we call $\tilde{u} = \nabla\psi^*(u)$, and take $u^* = \tilde{u} - \tilde{v}$, then

$$\psi^*(v) - \langle v, \tilde{u} \rangle + \psi(\tilde{u}) \geq \frac{1}{2L_{\psi^*}}\|\tilde{u} - \tilde{v}\|^2$$

and by Theorem 23.5 in Rockafellar, $\psi(\tilde{u}) = \langle u, \tilde{u} \rangle - \psi^*(\tilde{u})$, thus

$$\psi^*(v) - \psi^*(u) - \langle \tilde{u}, v - u \rangle \geq \frac{1}{2L_{\psi^*}} \|\tilde{u} - \tilde{v}\|^2$$

which proves the claim. $\qquad\square$

*Proof of Lemma 2.* We start by bounding the difference in Bregman divergences

$$
\begin{aligned}
D_{\psi^*}&(z^{(k+1)}, z^\star) - D_{\psi^*}(z^{(k)}, z^\star)\\
&= -D_{\psi^*}(z^{(k)}, z^{(k+1)}) + \left\langle \nabla\psi^*(z^{(k+1)}) - \nabla\psi^*(z^\star), z^{(k+1)} - z^{(k)} \right\rangle \quad \text{By Lemma 5}\\
&\leq -\frac{1}{2L_{\psi^*}} \|\tilde{z}^{(k+1)} - \tilde{z}^{(k)}\|^2 + \left\langle \tilde{z}^{(k+1)} - x^\star, -\frac{ks}{r}\nabla f(x^{(k+1)}) \right\rangle \qquad \text{by Lemma 6.}
\end{aligned}
$$
$$\tag{8}$$

Now using the step from $x^{(k+1)}$ to $\tilde{x}^{(k+1)}$, we have

$$\tilde{x}^{(k+1)} = \underset{x \in E}{\arg\min} \left\langle \nabla f(x^{(k+1)}), x \right\rangle + \frac{1}{\gamma s} R(x, x^{(k+1)})$$

with $\frac{\ell_R}{2}\|x - y\|^2 \leq R(x, y) \leq \frac{L_R}{2}\|x - y\|^2$. Therefore, for any $x$, $R(x, x^{(k+1)}) \geq R(\tilde{x}^{(k+1)}, x^{(k+1)}) + \gamma s \left\langle \nabla f(x^{(k+1)}), \tilde{x}^{(k+1)} - x \right\rangle$. We can write

$$\tilde{z}^{(k+1)} - \tilde{z}^{(k)} = \frac{1}{\lambda_k}\left(\lambda_k \tilde{z}^{(k+1)} + (1-\lambda_k)\tilde{x}^{(k)} - x^{(k+1)}\right) = \frac{1}{\lambda_k}\left(d^{(k+1)} - x^{(k+1)}\right),$$

where we have defined $d^{(k+1)}$ in the obvious way. Thus

$$
\begin{aligned}
\|\tilde{z}^{(k+1)} &- \tilde{z}^{(k)}\|^2\\
&= \frac{1}{\lambda_k^2}\|d^{(k+1)} - x^{(k+1)}\|^2\\
&\geq \frac{1}{\lambda_k^2}\frac{2}{L_R}R(d^{(k+1)}, x^{(k+1)})\\
&\geq \frac{1}{\lambda_k^2}\frac{2}{L_R}\left(R(\tilde{x}^{(k+1)}, x^{(k+1)}) + \gamma s\left\langle \nabla f(x^{(k+1)}), \tilde{x}^{(k+1)} - d^{(k+1)} \right\rangle\right)\\
&\geq \frac{1}{\lambda_k^2}\frac{2}{L_R}\left(\frac{\ell_R}{2}\|\tilde{x}^{(k+1)} - x^{(k+1)}\|^2 + \gamma s\left\langle \nabla f(x^{(k+1)}), \tilde{x}^{(k+1)} - \lambda_k \tilde{z}^{(k+1)} - (1-\lambda_k)\tilde{x}^{(k)} \right\rangle\right).
\end{aligned}
$$

Thus

$$
\begin{aligned}
\lambda_k \frac{kL_R}{2r\gamma}\|\tilde{z}^{(k+1)} - \tilde{z}^{(k)}\|^2 &\geq \frac{k\ell_R}{2r\lambda_k\gamma}\|\tilde{x}^{(k+1)} - x^{(k+1)}\|^2\\
&+ \left\langle \frac{ks}{r}\nabla f(x^{(k+1)}), \frac{1}{\lambda_k}\tilde{x}^{(k+1)} - \tilde{z}^{(k+1)} - \frac{1-\lambda_k}{\lambda_k}\tilde{x}^{(k)} \right\rangle.
\end{aligned}
$$
$$\tag{9}$$

Subtracting (9) from (8),

$$
\begin{aligned}
D_{\psi^*}&(z^{(k+1)}, z^\star) - D_{\psi^*}(z^{(k)}, z^\star)\\
&\leq -\alpha_k\|\tilde{z}^{(k+1)} - \tilde{z}^{(k)}\|^2 - \frac{k\ell_R}{2r\lambda_k\gamma}\|\tilde{x}^{(k+1)} - x^{(k+1)}\|^2\\
&+ \left\langle -\frac{ks}{r}\nabla f(x^{(k+1)}), -x^\star + \frac{1}{\lambda_k}\tilde{x}^{(k+1)} - \frac{1-\lambda_k}{\lambda_k}\tilde{x}^{(k)} \right\rangle,
\end{aligned}
$$

where

$$\alpha_k = \frac{1}{2L_{\psi^*}} - \frac{k\lambda_k L_R}{2r\gamma}.$$

Defining $D_1^{(k+1)} = \|\tilde{x}^{(k+1)} - x^{(k+1)}\|^2$ and $D_2^{(k+1)} = \|\tilde{z}^{(k+1)} - \tilde{z}^{(k)}\|^2$, we can rewrite the last inequality as

$$D_{\psi^*}(z^{(k+1)}, z^\star) - D_{\psi^*}(z^{(k)}, z^\star)$$
$$= -\alpha_k D_2^{(k+1)} - \frac{k\ell_R}{2r\lambda_k\gamma} D_1^{(k+1)} + \frac{sk}{r}\left\langle -\nabla f(x^{(k+1)}), \tilde{x}^{(k+1)} - x^\star \right\rangle$$
$$+ \frac{1-\lambda_k}{\lambda_k}\frac{sk}{r}\left\langle -\nabla f(x^{(k+1)}), \tilde{x}^{(k+1)} - \tilde{x}^{(k)} \right\rangle$$

By Lemma 4, we can bound the inner products as follows

$$\left\langle \tilde{x}^{(k+1)} - \tilde{x}^{(k)}, -\nabla f(x^{(k+1)}) \right\rangle \le f(\tilde{x}^{(k)}) - f(\tilde{x}^{(k+1)}) + \frac{L_f}{2} D_1^{(k+1)},$$
$$\left\langle \tilde{x}^{(k+1)} - x^\star, -\nabla f(x^{(k+1)}) \right\rangle \le f^* - f(\tilde{x}^{(k+1)}) + \frac{L_f}{2} D_1^{(k+1)}.$$

Combining these inequalities, and using the fact that $\frac{1-\lambda_k}{\lambda_k} = \frac{k}{r}$, we have

$$D_{\psi^*}(z^{(k+1)}, z^\star) - D_{\psi^*}(z^{(k)}, z^\star)$$
$$\le -\alpha_k D_2^{(k+1)} + \frac{k^2 s}{r^2}\left( f(\tilde{x}^{(k)}) - f(\tilde{x}^{(k+1)}) + \frac{L_f}{2} D_1^{(k+1)} \right) + \frac{ks}{r}\left( f^* - f(\tilde{x}^{(k+1)}) + \frac{L_f}{2} D_1^{(k+1)} \right)$$
$$- \frac{k\ell_R}{2r\lambda_k\gamma} D_1^{(k+1)}$$
$$= \frac{k^2 s}{r^2}\left( f(\tilde{x}^{(k)}) - f(\tilde{x}^{(k+1)}) \right) + \frac{ks}{r}\left( f^* - f(\tilde{x}^{(k+1)}) \right) - \alpha_k D_2^{(k+1)} - \beta_k D_1^{(k+1)},$$

where

$$\beta_k = \frac{k\ell_R}{2r\lambda_k\gamma} - \frac{L_f k^2 s}{2r^2} - \frac{L_f ks}{2r}.$$

Finally, we obtain a bound on the difference $\tilde{E}^{(k+1)} - \tilde{E}^{(k)}$

$$\tilde{E}^{(k+1)} - \tilde{E}^{(k)}$$
$$= \frac{(k+1)^2 s}{r}(f(\tilde{x}^{(k+1)}) - f^\star) - \frac{k^2 s}{r}(f(\tilde{x}^{(k)}) - f^\star) + r(D_{\psi^*}(z^{(k+1)}, z^\star) - D_{\psi^*}(z^{(k)}, z^\star))$$
$$= \frac{k^2 s}{r}(f(\tilde{x}^{(k+1)}) - f(\tilde{x}^{(k)})) + \frac{(2k+1)s}{r}(f(\tilde{x}^{(k+1)}) - f^\star) + r(D_{\psi^*}(z^{(k+1)}, z^\star) - D_{\psi^*}(z^{(k)}, z^\star))$$
$$\le \frac{(2k+1-kr)s}{r}(f(\tilde{x}^{(k+1)}) - f^\star) - r\alpha_k D_2^{(k+1)} - r\beta_k D_1^{(k+1)}$$

For the desired inequality to hold, it suffices that $\alpha_k, \beta_k \ge 0$, i.e.

$$\frac{1}{2L_{\psi^*}} - \frac{kL_R}{2(r+k)\gamma} \ge 0$$
$$\frac{k(r+k)\ell_R}{2r^2\gamma} - \frac{L_f k^2 s}{2r^2} - \frac{L_f ks}{2r} \ge 0,$$

i.e.

$$\gamma \ge \frac{kr}{kr + r^2} L_R L_{\psi^*}$$
$$s \le \frac{\ell_R}{L_f \gamma}.$$

So it is sufficient that

$$\gamma \ge L_R L_{\psi^*} \qquad\qquad s \le \frac{\ell_R}{L_f \gamma}$$

which concludes the proof. $\qquad\qquad\qquad\qquad\qquad\qquad\qquad\qquad\qquad\qquad\qquad$ $\square$

# 5    Bounding $\tilde{E}^{(1)}$

Here we derive the bound on $\tilde{E}^{(1)}$ that is used in Theorem 3. Suppose the assumptions of Theorem 3 hold. Then by Lemma 2, we have

$$\tilde{E}^{(1)} \leq \tilde{E}^{(0)} + \frac{s}{r}(f(\tilde{x}^{(1)})) - f^\star$$
$$= rD_{\psi^*}(z^{(0)}, z^\star) + \frac{s}{r}(f(\tilde{x}^{(1)}) - f^\star)$$

and we bound $f(\tilde{x}^{(1)}) - f^\star$. By definition, $\tilde{x}^{(1)} = \arg\min_{\tilde{x} \in E} \gamma s \left\langle \nabla f(x^{(1)}), \tilde{x} \right\rangle + R(\tilde{x}, x^{(1)})$ thus

$$\gamma s \left\langle \nabla f(x^{(1)}), \tilde{x}^{(1)} \right\rangle + R(\tilde{x}^{(1)}, x^{(1)}) \leq \gamma s \left\langle \nabla f(x^{(1)}), x^{(1)} \right\rangle \qquad (10)$$

Therefore,

$$f(\tilde{x}^{(1)}) - f^\star$$
$$\leq \left\langle \nabla f(x^{(1)}), \tilde{x}^{(1)} - x^\star \right\rangle + \frac{L_f}{2}\|\tilde{x}^{(1)} - x^{(1)}\|^2 \qquad \text{by Lemma 4}$$
$$\leq \left\langle \nabla f(x^{(1)}), \tilde{x}^{(1)} - x^\star \right\rangle + \frac{L_f}{\ell_R}R(\tilde{x}^{(1)}, x^{(1)}) \qquad \text{by assumption on } R$$
$$\leq \left\langle \nabla f(x^{(1)}), \tilde{x}^{(1)} - x^\star \right\rangle + \frac{1}{\gamma s}R(\tilde{x}^{(1)}, x^{(1)}) - \frac{L_f}{\ell_R}R(\tilde{x}^{(1)}, x^{(1)}) \quad \text{using that } \frac{2L_f}{\ell_R} \leq \frac{1}{\gamma s}$$
$$\leq \left\langle \nabla f(x^{(1)}), x^{(1)} - x^\star \right\rangle - \frac{L_f}{\ell_R}R(\tilde{x}^{(1)}, x^{(1)}) \qquad \text{by (10)}$$
$$\leq f(x^{(1)}) - f^\star + \frac{L_f}{2}\|x^{(1)} - x^\star\|^2 - \frac{L_f}{\ell_R}R(\tilde{x}^{(1)}, x^{(1)}) \qquad \text{by Lemma 4}$$
$$\leq f(x^{(1)}) - f^\star$$

finally, observing that $x^{(1)} = x_0$, we have $f(\tilde{x}^{(1)}) - f^\star \leq f(x_0) - f^\star$, therefore

$$\tilde{E}^{(1)} \leq rD_{\psi^*}(z_0, z^\star) + \frac{s}{r}(f(x_0) - f^\star)$$

which proves the desired inequality.

# References

R.T. Rockafellar. *Convex Analysis*. Princeton University Press, 1970.