[Reviews · NeurIPS 2015]

Submitted by Assigned_Reviewer_1

The paper combines the ODE interpretation of mirror descent with the work of [25], which interprets AGD as a discretized ODE, to formulate an ODE system that corresponds to what should be accelerated mirror descent. Proofs of existence, uniqueness, and convergence follow standard ODE methods using compactness and contraction mappings. Discretizing the ODE yields the derivation of discrete-time accelerated mirror descent methods, one example of which is accelerated entropic descent. The authors provide experiments validating the improvement of accelerated entropic descent over its non-accelerated variant.

Quality: The paper is relatively well-written, outlining recent work in the field, motivating its contribution, and methodically arriving at concrete algorithms at the end.

Clarity: The authors mention and leverage the Cauchy-Lipschitz theorem many times throughout the paper, but not once provide an exact statement or a reference. Since the theorem is sometimes presented in different forms, it seems important to cite it precisely. It is not clear why the 2nd inequality on line 251 is true. It is also not clear why the 2nd inequality in the proof of Lemma 2 on page 5 of the supplementary material is true. [15] is cited improperly without any journal or conference reference.

EDIT: The authors addressed many of these complaints in their rebuttal.

Originality: The formulation of accelerated mirror descent is a natural extension of recent work, but to the best of the reviewer's knowledge, new.

Significance: The results are new, but the proof techniques and ideas are relatively standard extensions to what exists in the literature. The experiments seem to indicate that the algorithms present an improvement upon commonly used methods.

Summary: The paper generalizes recent work on accelerated gradient descent in continuous time to the mirror descent framework and presents a natural contribution to the optimization literature. However, some of the presentation and arguments are unclear and would do well to be cleaned up.

Submitted by Assigned_Reviewer_2

This paper proposes a new accelerated mirror descent algorithm. The derivation of the algorithm is based on a careful analysis of the continuous time dynamics with subsequent discretization step.

The disconnect between contionous time dynamics and discrete time dynamics is a slight concern here. In particular, for continuous time dynamics one immediately gets O(1/k^2) convergence rate. However, discretization of this ODE system does not result in the system with the same convergence rate and discrete schemes needs to be modified (via additional proximal step) to achieve the same convergence rate. However, it is not immediately clear what is the relation between the original continuous dynamics and modified discrete scheme. Is there a continuous analogue of the modified discrete scheme?

Meta-concern: Optimization is of great importance in learning and hence of great interest to NIPS community, but perphaps optimization journal or conference would be a better fit for this work which is rather general and applies to settings other than machine learning. It would be nice if the authors would demostrate the relevance of this work to machine learning at least in experiments (in addition to solving a toy numerical example).
Summary: This paper proposes a new accelerated mirror descent algorithm. The derivation of the algorithm is based on a careful analysis of the continuous time dynamics with subsequent discretization step.

Submitted by Assigned_Reviewer_3

The authors propose a hybrid of mirror descent and Nesterov's accelerated method, motivated by a variant of the ODE

corresponding to the dynamical system implemented by Nesterov's accelerated method. The contribution is generally well-motivated and clearly-written and establishes a new family of algorithms that blend mirror descent and Nesterov's accelerated method.

Minor comments:

p. 2:

"consider the Lyapunov function...": for ease of readability, I suggest giving a reference to, or providing the definition of a Lyapunov function and briefly explaining its significance.

p. 4:, Lemma 1 {\cal I} in the proof is not defined. Also, please provide a citation for the Arzela-Ascoli theorem.
Summary: The manuscript is generally well-written and motivated, with a worthwhile contribution to an active area in the ML and optimization research literature.

Submitted by Assigned_Reviewer_4

The paper generalizes a recent ODE explanation of accelerated gradient descent in parallel to the generalization from GD to mirror descent. Discretization produces a set of accelerated mirror descent algorithms with 1/t^2 rates.

The paper clearly describes an approach to design and analysis of optimization algorithms, thus has a pedagogic contribution, in addition to the concrete resuting algorithms. The basic analysis and some elements are inherited from reference [25], so originality is not the highest, but there are certainly new algorithms and results. While an accelerated mirror descent existed (Zhu and Orrecia), having relatively simple and flexible derivations can lead to better adaptations to new settings.

Detailed comments: - Use of V(t) and V(k) is a bit confusing; the functions are very similar in terms of the corresponding x, but those x differ. The two distinct V(k) exacerbate it. - Give a reference or definition for "mixed forward/backward Euler scheme"
Summary: An excellent paper, giving a (relatively) simple development of accelerated mirror descent.

Author Feedback
Author rebuttal: We thank the reviewers for their assessment of the article and their constructive comments. We will address the concerns raised by some of the reviewers below.

The main concern seems to be the extent of the connection between the continuous-time and discrete-time versions (as raised by reviewers 2 and 7).

Two discretizations are considered: the first one is intuitive and shown for illustrative purposes, but inconclusive (in the sense that we could not proved the O(1/k^2) convergence rate), the second one uses an additional prox step to obtain the same O(1/k^2) rate. We showed (Section 4.2) that even with the additional prox step, the discrete system is consistent with the ODE, in the following sense: as the step size s tends to 0, the difference equations converge to the continuous-time differential equations. This establishes a strong connection between the continuous-time and discrete-time systems.
In addition, the Lyapunov functions used in both cases are identical, up to a change in their argument: the continuous-time version can be written as $V(t) = g(X(t), Z(t))$, and the discrete version is $V(k) = g(\tilde x^{(k)}, z^{(k)})$), where $g$ is the same function in both cases.

To summarize, the extent of the connection is as follows:
1) The discrete systems (both versions) are consistent with the continuous-time ODE.
2) The same O(1/k^2) convergence rate is obtained for the second discretization.
3) The discrete and continuous Lyapunov functions are identical, up to a change in the argument.

It is also worth pointing out that both the continuous and discrete systems can be used for constrained optimization (this will be made more explicit in the revised version). As long as $\nabla \psi$ maps onto the convex set $E$, the trajectories (continuous and discrete) remain in $E$. This was only proved in the discrete case (line 244), however, we realized after the initial submission that a similar argument can be made in the continuous case: the equation $\dot X = -\frac{t}{r} (\nabla \psi(Z) - X)$ can be written in its integral form as $X = \frac{\int_0^t \tau^{r-1} \nabla \psi(Z(\tau)) d\tau}{\int_0^t \tau^{r-1} d\tau}$, thus by convexity of E, X remains in E.
Incidentally, this form of the ODE gives a new, compelling interpretation of the dynamics: the dual variable obeys $\dot Z = -\frac{t}{r} \nabla f(X)$, and the primal variable $X$ is simply the weighted average of the projections $\nabla \psi(Z(t))$, with increasing weights $t^{r-1}$. This also gives an interpretation of $r$ as a parameter controlling the weight distribution. We will add a brief discussion of this form of the ODE and its implications.

We agree with reviewer 2 that the results of the paper could be of interest beyond the machine learning community. At the same time, we believe that understanding and streamlining the design and analysis of first-order methods is of particular interest to the machine learning community, which makes NIPS a fitting venue for this work, and previous related work such as [6,7].

Missing references, typos and notation issues:
- Definition of a Lyapunov function (reviewer 4): we will give a formal definition and reference section 4.1 in [1].
- Cauchy-Lipschitz theorem (reviewer 1): we will add the following references: Theorem 2.5 (existence and uniqueness of a C^1 solution) and Theorem 2.8 (continuous dependence on initial conditions) in [2].
- Equation line 251 (reviewer 1): s should be replaced with \frac{ks}{r}. The equality follows from Theorem 23.5 in [3].
- Second-to-last equality page 5 in the supplementary file (reviewer 1): we will reference Proposition 11.1 in [4].
- Proof of Theorem 1: missing definition of \mathcal{I} (reviewer 4): these are the indices of the converging subsequence. This will be made clear.
- Missing reference for forward and backward Euler scheme (reviewer 3): we will reference Chapter 2 in [5].
- Confusing notation for the discrete Lyapunov functions (reviewer 3): we will use a different notation for the two discrete Lyapunov functions.
- Improper citation of reference [15]: this will be fixed.

We thank the reviewers again for their careful reading and their constructive comments.

[1] H.K. Khalil, Nonlinear systems, Macmillan Pub. Co., 1992.
[2] G. Teschl, Ordinary differential equations and dynamical systems, American Mathematical Society, 2012.
[3] T.R. Rockafellar, Convex Analysis, Princeton University Press, 1997.
[4] N. Cesa-Bianchi and G. Lugosi. Prediction, learning, and games. Cambridge University Press, 2006.
[5] J. Butcher, Numerical Methods for Ordinary Differential Equations, Second edition, John Wiley & Sons, 2008.
[6] W. Su, S. Boyd, and E. Candes. A differential equation for modeling nesterov's accelerated gradient method: Theory and insights. In NIPS, 2014.
[7] C. Wang, X. Chen, A. J. Smola, and E. P. Xing. Variance Reduction for Stochastic Gradient Optimization. In NIPS, 2013.